# Automatic Multiview Alignment of RGB-D Range Maps of Upper Limb Anatomy

**DOI:** 10.3390/s23187841

**Published:** 2023-09-12

**Authors:** Luca Di Angelo, Paolo Di Stefano, Emanuele Guardiani, Paolo Neri, Alessandro Paoli, Armando Viviano Razionale

**Affiliations:** 1Department of Industrial and Information Engineering and Economics, University of L’Aquila, 67100 L’Aquila, Italy; luca.diangelo@univaq.it (L.D.A.); paolo.distefano@univaq.it (P.D.S.); 2Department of Civil and Industrial Engineering, University of Pisa, 56122 Pisa, Italy; paolo.neri@unipi.it (P.N.); alessandro.paoli@unipi.it (A.P.); armando.viviano.razionale@unipi.it (A.V.R.)

**Keywords:** depth cameras, 3D optical scanning, upper limb anatomy, automatic point cloud alignment, neural network

## Abstract

Digital representations of anatomical parts are crucial for various biomedical applications. This paper presents an automatic alignment procedure for creating accurate 3D models of upper limb anatomy using a low-cost handheld 3D scanner. The goal is to overcome the challenges associated with forearm 3D scanning, such as needing multiple views, stability requirements, and optical undercuts. While bulky and expensive multi-camera systems have been used in previous research, this study explores the feasibility of using multiple consumer RGB-D sensors for scanning human anatomies. The proposed scanner comprises three Intel^®^ RealSenseTM D415 depth cameras assembled on a lightweight circular jig, enabling simultaneous acquisition from three viewpoints. To achieve automatic alignment, the paper introduces a procedure that extracts common key points between acquisitions deriving from different scanner poses. Relevant hand key points are detected using a neural network, which works on the RGB images captured by the depth cameras. A set of forearm key points is meanwhile identified by processing the acquired data through a specifically developed algorithm that seeks the forearm’s skeleton line. The alignment process involves automatic, rough 3D alignment and fine registration using an iterative-closest-point (ICP) algorithm expressly developed for this application. The proposed method was tested on forearm scans and compared the results obtained by a manual coarse alignment followed by an ICP algorithm for fine registration using commercial software. Deviations below 5 mm, with a mean value of 1.5 mm, were found. The obtained results are critically discussed and compared with the available implementations of published methods. The results demonstrate significant improvements to the state of the art and the potential of the proposed approach to accelerate the acquisition process and automatically register point clouds from different scanner poses without the intervention of skilled operators. This study contributes to developing effective upper limb rehabilitation frameworks and personalized biomedical applications by addressing these critical challenges.

## 1. Introduction

Digital representations of anatomical structures play a crucial role in various applications within the biomedical field. For example, an accurate 3D model depicting the anatomy of the upper limb can be utilized to design customized devices for personalized rehabilitation or human–machine interfaces (HMIs) [1]. Enhancing upper limb functionality holds immense importance for patients afflicted with sports injuries, strokes, or dystonia disorders, as the upper limb is extensively involved in daily activities. Novel technologies like virtual, augmented, and mixed reality offer promising solutions to overcome conventional treatment limitations [2]. Frameworks for upper limb rehabilitation can be defined by integrating virtual reality gaming and tailored controllers, enabling the recreation of therapeutic movements. Patients can enter a virtual gaming environment tailored to their rehabilitation needs and engage with the game and virtual environment using a specifically configured controller [3].

The initial and crucial step in generating a digital representation of an anatomical part involves 3D scanning [4]. In this regard, optical techniques prove to be particularly suitable as they cause minimal invasiveness to the patient [5]. However, 3D scanning of the forearm is a complex process due to the requirements of multiple scans from various angles and ensuring the arm remains stable to prevent motion artifacts caused by involuntary movements. Furthermore, the presence of optical undercuts complicates the acquisition of finger geometry. Numerous approaches are available for capturing human body parts. Photogrammetry is the underlying principle for 3D body scanners, which can simultaneously capture all sides of the human body. These systems employ multiple cameras arranged in a cylindrical architecture to capture the target anatomy from different viewpoints [6,7,8]. Nonetheless, these systems can be cumbersome and expensive, depending on the cameras used.

A practical alternative is given by handheld 3D scanners that can be manually moved around the patient [9,10]. This approach requires longer acquisition times due to the scanner’s need to cover the entire target anatomy [4]. A drawback of this method is the possibility of unintentional patient movement during the scanning process, resulting in artifacts in the scanning outcome. Furthermore, high-level handheld scanners, which offer substantial reductions in acquisition time compared with low-level handheld scanners, tend to be expensive [4,11]. For this reason, in the last few years, many studies have been conducted to assess the feasibility of using consumer RGB-Depth (RGB-D) sensors (e.g., Kinect by Microsoft, Structure by Occipital, RealSense by Intel) to scan human anatomies [12,13]. RGB-D cameras are compact, portable, and low-cost [4]. In this regard, some architectures have been developed to scan human arms, considering the simultaneous use of multiple RGB-D sensors rigidly assembled in circular jigs [14,15]. These solutions have the advantage of offering affordable architectures that can ensure the necessary dimensional accuracy.

However, rapidly obtaining the arm’s complete anatomy requires multiple sensors to capture acquisitions from various viewpoints, effectively resolving optical undercut issues. Consequently, this leads to the development of stationary, bulky, and non-portable systems [14]. In contrast, lightweight configurations can be achieved by integrating a limited number of sensors, although this approach necessitates multiple scanner placements around the patient’s arm to capture the complete anatomy [15]. This technique is more flexible and adaptable to different patients’ needs but requires accurate registration of the acquired multiple views into a common reference frame, which is conventionally composed of two alignment steps [16,17]: a local coarse alignment and a global fine registration. The first step is based on manually identifying common landmarks on different scans. While widely used, the traditional manual procedure has several limitations that negatively impact performance. The manual selection of corresponding points is time-consuming and subjective, relying on the user’s visual inspection and interpretation of the acquired data. This can lead to different results depending on the operator, making the procedure unreliable and non-repeatable. Furthermore, the absence of markers on the patient’s arm complicates the process: the operator must rely solely on visual inspection to identify corresponding points, leading to errors and inconsistencies. The global fine registration, instead, is based on applying iterative closest point (ICP) techniques on overlapping regions of adjacent scans. Aligning 3D human body scans captured from different viewpoints presents challenges, especially when there is low overlap and a lack of relevant topology, as is often the case with the human forearm, which typically exhibits a quasi-cylindrical shape.

To overcome these limitations, this paper proposes an automatic alignment procedure for human forearm scans from a low-cost handheld scanner. The scanner consists of three Intel^®^ RealSenseTM D415 depth cameras rigidly assembled on a lightweight circular jig, enabling the simultaneous acquisition of point clouds from three different viewpoints. Extrinsic calibration is performed to determine the positioning and orientation of the three depth sensors, aligning the three point clouds into a common reference frame for each scanner pose.

The primary contribution of this research lies in providing an automated procedure to align scans obtained from different scanner poses, where extrinsic calibration cannot be used due to the operator moving the scanner around the target anatomy. Regarding registration methods, feature-based approaches are commonly employed to identify meaningful and robust geometrical descriptors, which serve as correspondences to estimate transformations between distinct point clouds [18]. These features can be artificially attached to the patient’s skin as markers or manually identified in the acquired point clouds during post-processing. However, both approaches are ineffective in clinical scenarios. Attaching markers to the patient’s skin is generally not feasible, and relying on manual intervention during post-processing requires skilled operators, which undermines the goal of automation.

Consequently, it is necessary to automatically extract local and/or global shape descriptors by analyzing the geometric information in the point clouds [19,20]. It is important to note that the accuracy of extracting these shape descriptors improves with high-density point clouds due to their reliance on discrete differential geometry. Nevertheless, the resolution of the depth cameras used in the 3D scanner must provide more information for the abovementioned methods to be utilized. As a result, an automatic procedure has been developed in the present work to identify common key points between scans obtained from different scanner placements. RGB images from the depth cameras are processed using a neural network to detect a dataset of relevant key points on the hand, which are then projected onto the point cloud to identify corresponding 3D points. A smaller dataset of relevant key points on the forearm is also included in the analysis. This additional dataset is generated by directly processing the point cloud with a specially developed algorithm that searches for matches along the forearm’s skeleton line. These two datasets, obtained from different scanner positions, are then used to achieve an automatic rough 3D alignment of the point clouds in a common reference frame. Finally, an iterative-closest-point (ICP) algorithm, supported by a specially designed function to be minimized, refines the alignment through fine registration.

The method was tested on the alignment of upper limb anatomies with multiple views for five subjects acquired with a low-cost handheld 3D scanner. The results are critically discussed by using as a reference a gold-standard manual alignment pipeline with commercial software and compared with the available implementations of published methods. The study aims to address two significant issues that arise when obtaining a 360° acquisition of an upper limb: speeding up the acquisition process and automatically registering point clouds derived from different scanner poses.

## 2. Related Works

Registering partially overlapped point clouds presents challenges due to limited and blurred information caused by occlusion, missing points, outliers, and measurement noise. According to [21], the existing methods in the related literature can be categorized into two main groups:Optimization-based methods;Learning-based methods.

### 2.1. Optimization-Based Registration Methods

These methods, which are typically applied to a pair of point clouds, generally involve two main steps:Estimating correspondences between the two point clouds to achieve a coarse-registration solution;Estimating the rigid body transformation by solving a least-squares problem to obtain the final solution.

A coarse estimation can be performed by extracting local 3D descriptors. These descriptors typically calculate statistics related to the principal component analysis (PCA) or other discrete geometric properties, such as surface normal, curvatures, second-order moment invariants, etc. A comprehensive review of 3D point cloud descriptors is provided by Han et al. [22]. However, these descriptors often fail to provide a complete and unambiguous representation of local shape geometry. Moreover, uncertainties arise due to factors like point cloud quality, resolution, surface noise, and outliers, further impacting the accuracy of these geometric properties. These challenges become even more pronounced when dealing with partially overlapping point cloud registration. To address these issues, the published methods for registering partially overlapping point clouds often rely on randomized alignment or overlap ratio estimation [23]. Two widely used randomized alignment-based techniques are four-point congruent sets (4PCS) [24] and Super4PCS [25]. These methods exploit the invariance of the ratio between lines formed by four coplanar points, eliminating the need for complex geometric calculations. The robustness of these algorithms increases with larger overlap areas compared with the average density of the point cloud, making them more noise-resilient.

The iterative closest point (ICP) [26] algorithm is commonly used for fine registration; however, it is prone to local optima, is unable to differentiate between inlier and outlier correspondences, and performs global registration that considers non-overlapping areas. Therefore, specific methods have been developed to enhance registration performance for partial overlap problems. The trimmed ICP method, for example, introduced overlap-ratio estimation [27,28] to exclude outliers and increase the algorithm’s robustness when analyzing point clouds with a relatively low overlap ratio (below 50%). Chetverikov et al. [27] proposed the trimmed ICP (TrICP) algorithm, which exploits the least-trimmed-squares approach to minimize a certain number of smaller values and sort the square errors. This method estimates the optimal transformation at each iteration step, proving high robustness for incomplete and noisy data with overlap ranges from 0.4 to 1. To reduce false correspondences and enhance robustness against outliers, Du et al. [29] introduced a new objective function that considers the overlapping percentage between the two point sets. Dong et al. [30] proposed the LieTrICP algorithm, which, after determining the correspondences, estimates the transformation by minimizing the trimmed squared distances of the point pairs using the Lie group parametrization. Incorporating the Lie group parametrization allows the algorithm to handle data with anisotropic scale transformation and low overlap rates [31]. Additionally, Wang et al. [28] proposed the parallel-trimming iterative-closest-point (PTrICP) method for fine point cloud registration. PTrICP incorporates parallel overlap rate estimation during the iterative registration process to improve the algorithm’s robustness. However, it is worth noting that all of these registration algorithms require a good initial estimation for satisfactory results [32].

### 2.2. Learning-Based Registration Methods

Learning-based registration methods use learning of the latent representations of the point clouds for their registration.

The first published method applying deep learning to point cloud registration is PointNetLK [33]. It uses a learnable global representation for point clouds based on PointNet [34]. This method’s main contribution is modifying the Lucas and Kanade (LK) algorithm [35] for optimization. Specifically, it avoids the need for convolutions on the PointNet representation. Sarode et al. [36] proposed a similar approach by replacing the LK algorithm with a multi-layer perceptron to improve noise robustness. One limitation of these methods is that they do not consider the adverse effects caused by non-overlapping regions. To overcome this limitation, Zhou et al. [37] proposed SCANet, which consists of two main modules: a spatial self-attention aggregation (SSA) module and a channel cross-attention regression (CCR) module. The SSA module efficiently extracts different-level features from point clouds to capture local and global information. The CCR module was innovatively used for pose estimation by enhancing relevant information and suppressing redundant information. The results of SCANet show a significant improvement in registration accuracy compared with state-of-the-art methods while reducing the number of weights. Xu et al. [38] proposed OMNet, another global-feature-based registration network robust to noise and different partial overlaps. OMNet uses learning masks to reject non-overlapping regions and allows iterative mask prediction and transformation estimation modification. However, estimating the masks without interaction with feature information is challenging, as pointed out in [39]. To address this challenge, Zhang et al. [39] proposed a two-stage partially overlapping point cloud registration method, which can deal with point clouds having inhomogeneous density. It uses a network with overlapping-region-prediction and pose-estimation modules. The first module extracts features from the two point clouds using the DGCNN algorithm [40] to predict possible overlapping regions. The second module combines PointNet with a self-attention mechanism to capture global information and achieve robust transformation regression. When tested on different datasets, this method proved to have limited accuracy compared with other methods, leading the authors to conclude that its results should be used as the initial pose for ICP.

In recent years, several research papers have introduced graph-based learning methods to process point clouds [40,41,42,43,44,45,46,47]. Unlike point-based methods, these approaches do not directly handle individual points as input. Instead, they construct local structures using information obtained from 3D shapers, treating them as graphs. Shen et al. proposed KCNet [42], which extracts local structural features using kernel correlation layers. Wang et al. [40] introduced EdgeConv, a novel operation for learning from point clouds, aiming to capture local geometric features while maintaining permutation invariance. Their reported results demonstrated that the model could learn to semantically group points by dynamically updating a relationship graph across layers. Zhang et al. [41] introduced LDGCNN, which draws inspiration from PointNet [34] and DGCNN [40], to construct a directed graph for point cloud data. The constructed graph extracts features, which are then utilized for point cloud classification and segmentation. The results indicate that LDGCNN reduces the network’s model size by eliminating the transformation network and optimizing the network architecture by connecting hierarchical features from various dynamic graphs. However, the main drawback of the graph-based methods mentioned above is the standard definition of convolution, which overlooks the differences between neighboring points. To address this issue, Wang et al. introduced a new graph attention convolution method called GAC in their study [44]. GAC utilizes learnable kernel shapes that can adapt dynamically to the object’s structure for point cloud semantic segmentation. Similarly, in another work, Wang et al. [32] presented STORM, a structure-based overlap-matching approach that employs differentiable sampling. This method predicts overlaps and generates partial correspondences for partial point cloud registration. The authors used densely connected EdgeConv layers [40] along with two transformers, demonstrating favorable outcomes, particularly in cases where the overlap ratio was low.

## 3. The Proposed Methodology

State-of-the-art analysis indicates that the morphology of the upper limb with automorphisms, combined with the small overlapping areas and high level of noise obtainable from different scans, makes applying existing optimization-based registration methods difficult. The same reasons, to which must be added the need for many training samples considering individual variations, make using learning-based methods unfeasible. To address these challenges, this research paper introduces a novel optimization-based method that focuses on registering anatomical features. The proposed approach, outlined in Figure 1, involves the following key steps:-Acquisition of raw 3D point clouds;-Detection of anatomical features:Hand key-point detection;Forearm key-point detection;-Coarse registration of the point clouds;-Fine registration of the point clouds.

The framework developed in the present work integrates a neural network to identify relevant key points on the patient’s hand with an automatic recognition process for forearm key points. Finally, a global registration by an ICP algorithm based on a specifically designed function to be minimized is carried out to refine the alignment results.

### 3.1. Acquisition of Raw 3D Point Clouds

The 3D scanner consists of three Intel^®^ RealSense^TM^ D415 depth cameras arranged at 90° angles on a circular frame (Figure 2a). The D415 sensor (Figure 2b) is a compact device (99 mm × 20 mm × 23 mm) equipped with an RGB camera (resolution up to 1920 × 1080 pixels, frame rate 30 fps), an infrared projector, and two IR cameras (resolution up to 1280 × 720 pixels, frame rate 90 fps). Unlike structured-light approaches that use sequential projection of fringe patterns (such as the SR300 Intel^®^ RealSense^TM^ depth camera), the infrared projector of the D415 sensor projects a single static pattern, thus avoiding interference issues among multiple sensors. The D415 camera has wide acquisition depth sensing, ranging from about 160 mm to 10,000 mm. However, as demonstrated in [48], the maximum accuracy is obtained for close distances (from 150 mm to 500 mm), with corresponding 3D reconstruction errors below 1 mm. In the present work, three depth cameras were arranged to acquire the target shape in the range 250 mm–450 mm in room ambient lighting conditions. The scanner generates three separate point clouds, automatically registered into a common reference frame through an extrinsic calibration process after the assembly stage. The extrinsic calibration, which solves the placement parameters between the depth cameras, uses a 3D calibration specimen and a coarse-to-fine scheme [49]. The calibration specimen consists of six differently oriented planar surfaces and three colored markers affixed on the external surfaces (Figure 2c). The scanner captures the specimen, thus providing three point clouds, one for each sensor. The marker’s centers on the captured RGB images are detected and reprojected on the corresponding point clouds using the sensor’s factory intrinsic calibration data for each sensor acquisition. The markers’ 3D coordinates are then used to obtain a coarse alignment of the point clouds, which is then refined by minimizing the distance between the point clouds through an iterative-closest-point (ICP) algorithm. The resulting roto-translation matrices, which transform each point cloud from the sensor reference frame to the common reference frame, are saved in a calibration file. This file aligns the three point clouds corresponding to a specific scanner pose during the 3D scanning of the patient’s arm. For more comprehensive information regarding the scanner’s architecture and the extrinsic calibration process, please refer to [50]. The scanner captures several consecutive frames of the limb to obtain a 360° scan of the patient’s arm. A user, such as a therapist, can hold the scanner with one or two hands and move it around the patient’s arm in a circular path (see Figure 2d).

The results of the acquisition process are (Figure 3):-N triplets {Τ_1_ (℘_1,1_, ℘_1,2_, ℘_1,3_), Τ_2_ (℘_2,1_, ℘_2,2_, ℘_2,3_), …, Τ_N_ (℘_N,1_, ℘_N,2_, ℘_N,3_)} of coarsely registered point clouds (Figure 3a);-3 × N images of the upper limb anatomy (Figure 3b).

The adopted depth-camera-based scanner makes the realignment of point clouds more challenging due to the low density of the point clouds and the high level of noise that characterizes the raw data.

### 3.2. Anatomical Features Detection

One of the most innovative aspects of the proposed method is achieving the initial alignment of N-point clouds through anatomical feature recognition. This operation is highly complex because it involves extracting high-level information from low-level information, such as the coordinates of the points, which are affected by noise and sparse distribution. To address this challenge, the proposed framework combines a neural network for identifying significant key points in 2D images of the patient’s hand with an automatic recognition process for key points on the forearm in the 3D point cloud.

#### 3.2.1. Hand Key-Point Detection by a Neural Network

Detecting hand key points using a neural network (NN) is a computer vision task that involves identifying specific landmarks on fingers, palms, and wrists. The use of NNs in studying hand motion has gained popularity due to their capacity to handle vast amounts of data and learn from it [51]. In the present study, the effectiveness of an NN in identifying landmarks on the patient’s hand is significantly enhanced by providing three RGB images for each scanner pose, thereby increasing the likelihood of recognizing relevant landmarks in at least one of the images. Additionally, various pre-trained detection models of hand key points are available as open-source software libraries, such as the OpenPose model [52] and the MediaPipe Hand Landmarker model [53]. These models can be fine-tuned using new data to enhance accuracy for specific tasks.

The MediaPipe Hand Landmarker model (v. 0.8.8.1), developed by the Google Research team, is used in this work. It is a component of the MediaPipe framework, which offers a flexible and scalable solution for real-time computer vision and machine learning tasks like hand tracking, facial recognition, and object detection. The MediaPipe Hand Landmarker model is specifically designed for real-time hand tracking and hand gesture recognition. It combines a deep neural network and computer vision algorithms to accurately detect and track hands, even in challenging environments like low light or cluttered backgrounds.

The neural network used in the MediaPipe Hand Landmarker model consists of multiple stages, including a hand segmentation network, a hand pose estimation network, and a hand landmark localization network. It outputs a set of 2D hand key points representing the locations of fingers, palms, and wrists. The MediaPipe Hand Landmarker model is implemented in Python, and its open-source code simplifies its integration into the overall scanning workflow, coded in Matlab.

When an RGB image of a hand is provided as input, the neural network returns a list of identified key points along with a recognition score. Under optimal conditions, the algorithm can identify 21 hand features, including four key points on each finger and one key point on the wrist. The output is always sorted in the same sequence of key points, ensuring consistent numbering, as shown in Figure 4. In cases where one or more key points are not visible in a particular image, the network assigns a “not-a-number” value to those undetected key points. As a result, the output always consists of 21 points in the same sequence, eliminating any confusion regarding landmark numbering. This data structure is crucial for comparing the same key points detected from different viewpoints, as it ensures consistent indexing and point correspondence.

Each acquisition frame includes data from three separate Intel^®^ RealSenseTM D415 sensors. The calibration of the scanner ensures that the three point clouds are aligned. Consequently, the corresponding RGB image for each point cloud can be analyzed to identify the hand key points. In theory, this process would provide three sets of hand key points that could be utilized for registering the clouds in subsequent frames. This redundancy in the data significantly increases the success rate. However, despite the hand’s orientation to the cameras and finger pose, some sensors may fail to detect the hand key points due to occlusion. Using a single sensor may result in a lower number of identified key points or complete failure. Therefore, an automated algorithm must assess each frame and select the sensor with the highest detection rate among the three options. It is important to note that cloud registration requires a minimum of three points; thus, detecting all 21 key points is unnecessary. However, more key points lead to better performance by improving registration constraints. When a hand pose does not allow the detection of at least three key points in any of the three images, the automatic coarse alignment fails, and an operator must intervene to align the relative scan to the previous ones. A metric was established to determine the quality of key-point detection by multiplying the identification score (provided by the NN algorithm for each RGB image) by the number of detected key points. This criterion allows the selection of the key-point group with the most landmarks and the best identification among the three sensors. This criterion allows selection of the key-point group with the most landmarks and the best identification among the three sensors. Once the best 2D key-point set is chosen, its coordinates can be projected onto the 3D point cloud using the D415 factory calibration. This results in a set of 3D key points that can be stored alongside the point cloud for registration purposes. An overview of the automatic selection method is depicted in Figure 5. It should be noted that if none of the sensors provide reliable key-point identification for a specific pose, manual registration will be necessary.

This process generates a set of 3D key points for each point cloud. However, due to the results obtained from the neural network (NN), the key points associated with each frame may consist of different landmarks. This poses a challenge, because cloud alignment requires two sets of corresponding 3D points. Typically, the NN may fail to identify a few random key points on the hand, which limits the usability of the entire set for rough registration. To overcome this issue, the intersection of the key-point lists from the two point clouds to be registered is considered. A list of corresponding indexes is stored alongside the point cloud. Only the key points with the selected indexes are used to determine the alignment roto-translation during the registration step.

#### 3.2.2. Forearm Key-Point Detection

To detect the forearm key points, each triplet captured by the three depth cameras and registered using the method proposed in Section 3.1 undergoes the following steps:-Initial evaluation of the approximate skeleton line of the forearm;-Final evaluation of the approximate skeleton line of the forearm;-Detection of forearm key points.

Let Τ_i_{℘_i,1_, ℘_i,2_, ℘_i,3_} represent the i-th triplet corresponding to the i-th scanner pose (Figure 6a). The direction of the first-attempt skeleton line of the forearm (**ξ**_o_) is calculated using the point cloud principal component of inertia: the longitudinal axis **ξ**_o_ is associated with a lower moment of intertia of T_i_. In Figure 6b, T_i_ is transformed so that ξ_o_ aligns with the *z*-axis of the global reference system and the origin (O) with the barycenter (avg_i-th_) of the hand key points detected for T_i_. Starting from ξ_o_, the final direction ξ_f_ is obtained by following these steps:-Introducing a set of planes (Π) perpendicular to the *z*-axis with a defined pitch and intersecting T_i_ from a fixed height below the barycenter (Figure 6c);-Clustering the points (Γ_k_) obtained for each plane intersection Π_k_ (Figure 6c);-Approximating each Γ_k_ with an ellipse whose center is **C**_k_ (Figure 6d);-Approximating the centers **C**_k_ with a line by the RANSAC algorithm, thus obtaining **ξ**_f_ (Figure 6e).

The forearm key points (FKP_h_) are finally identified as the intersection points between ξ_f_ and the set of planes perpendicular to ξ_f_, which start from a fixed height below the barycenter (avg_*i*-th_) (Figure 6f).

### 3.3. Coarse Registration of Point Cloud

Figure 7 shows the outcome of registration performed after the acquisition stage and recognition of key points for the hand and forearm in the six triplets illustrated in Figure 3. Additionally, the same figure presents cross-sections at various distances to the barycenter of the identified hand markers:

z = 25 (fingers);

z = −100 (wrist);

z = −250 (forearm).

The registration quality is influenced by extrinsic calibration as described in Section 3.1.

As a first step, the coarse-registration algorithm selects the triplet with the highest number of recognized hand key points as the fixed reference (T_f_) cloud. Subsequently, all other triplets are registered to this fixed triplet using their corresponding key points on the hand and forearm. To achieve this, the roto-translation matrix is obtained by solving the weighted absolute orientation problem using Horn’s quaternion-based method [54]. Figure 8 shows the result of the coarse registration of the six triplets reported in Figure 3.

### 3.4. Fine Registration of Point Clouds

The final step of the proposed framework involves the simultaneous fine registration of all the *T_i_* point clouds. For this purpose, an original automated procedure, summarized in the following, has been coded into a Python application.

One of the original elements of this approach is the method implemented for the fine registration of two point clouds (℘*_fixed_* and ℘*_moving_*), which finds the rigid rotation [*R*^opt^] and translation [t^opt^] such that:(1)$$\left[R^{opt},t^{opt}\right]=\arg\underset{R,t}{\min}\left\|\frac{1}{f_{1}f_{2}}\right\|$$

The *f_1_* term in the objective function evaluates the number of points in the clouds whose distance is lower than a specified threshold value, *toll*. It is defined as:(2)$$f_{1}=\frac{|D|\;}{min(M,N)}$$
where:

*M* = number of points of ℘_fixed_

*N* = number of points of ℘_moving_

*D* is a *M* × *N* matrix, defined as follows:(3)$$D_{MxN}=\left\{\begin{array}{l} d_{ij}=distance\left(P_{i}\in {℘}_{fixed,j}\in {℘}_{aligned}\right)\;\\ d_{ij}=NULL\; \end{array}\right.\begin{array}{c} if\;distance<toll\\ otherwise \end{array}$$
whose *d_ij_* is a no-null element representing the Euclidian distance between the *i-th* point of ℘_fixed_ and the *j*-th point of ℘_moving_ if it is lower than the *toll* threshold value.

The term *f_2_* refines the angular mismatch between the two point clouds and is defined as:(4)$$f_{2}=\frac{\sum_{\begin{array}{c} 0\leq \;i\;\leq \;M\\ 0\leq j\leq N\; \end{array}}v_{ij}\;}{\left|D\right|}$$
where $v_{ij}$ are the elements of the V_M × N_ matrix, determined as:(5)$$V_{M\times N}=\left\{\begin{array}{l} v_{ij}=\hat{n}(P_{i})∙\hat{n}(Q_{j})\;\\ v_{ij}=NULL\; \end{array}\right.\begin{array}{c} if\;d_{ij}<toll\\ otherwise \end{array}$$
with $\hat{n}(P)$ being the exterior normal estimated for the point *P*.

The constrained nonlinear optimization problem resulting from the proposed fine registration of two point clouds is solved using a Python implementation of the sequential least-squares programming method [55]. This optimization method was selected for its efficiency in solving constrained nonlinear optimization problems subject to the lower and upper bounds values of angle rotation and distance translation. During the preliminary experimental analysis, it was noticed that there was a significant impact of outliers resulting from the depth camera acquisitions on the results of the fine-registration process. In this specific application, two kinds of outliers can be identified in the scans: a first one of random nature (noise) and a second one of systematic origin, identifiable in the peripheral areas of the hand and the arm. Both of these outliers were removed from the original datasets by applying an SOR filter implemented in Open3D (see Figure 9 for details).

The second innovative aspect of the global fine registration involves a strategy to realign all the point clouds {℘_1_, ℘_2_, …,℘_np_} using the optimization method described above for two point clouds. This strategy is summarized in Algorithm 1. Figure 10 shows the results of the fine registration of the six triplets reported in Figure 3, highlighting a significant improvement to the coarse registration shown in Figure 8.
**Algorithm 1.** Developed algorithm for the fine global registration of np point clouds.**Require:**Point clouds {℘10, ℘20, …,℘np0}, N_iter_ex_Labeling of point clouds from 1 to np in a clockwise direction:***
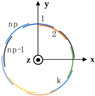
***for h = 1 to N_iter_ex_ do  Select randomly the index of the fixed point cloud:  Index_fixed_point_cloud = random(1,np)  Reassign the labels to point clouds such that:  -Index_fixed_point_cloud ← 1-Labels from 2 to np in a clockwise direction
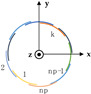
   for k = 2 to np do     Rk,tk = fine_registration(℘_fixed,_ ℘kh−1)     Update the ℘_k_ alligment:     ℘kh=Rk∙℘kh−1+tk     Update the ℘_fixed_:     ℘fixed⟵℘fixed⋃ ℘kh   end forend for

## 4. Results and Discussion

The computer-based methodology proposed in this study was implemented using custom software coded in MATLAB 2022. This decision allowed for the definition of a unified framework for automatically realigning point clouds by incorporating functions written in Python. Table 1 presents the parameter values used to implement various steps within the proposed procedure, as applied in the case studies.

This study evaluates the proposed methodology’s potential for realigning morphologically complex shapes, specifically focusing on human upper limbs. The arms of healthy subjects (five volunteers from the Department of Civil and Industrial Engineering at the University of Pisa) were used to ensure that the quality of the overall acquisition was not influenced by pathologies such as strokes or dystonia disorders.

### 4.1. Assessment of the Performance of the Proposed Methodology

Firstly, to quantitatively assess the performance of the proposed methodology, the obtained final registration results were compared with those obtained by a gold-standard approach: manual coarse alignment of the point clouds by an operator followed by an ICP algorithm for fine registration (Algorithm 2) using commercial software (i.e., Geomagic Studio 2014^®^ v. 1.0 (from 3D Systems, Rock Hill, SC, USA). The resulting point clouds were transformed into a common reference system to enable a fair comparison between the automatic and manual methods. In this reference system, the origins correspond to the respective centers of gravity of the point clouds. At the same time, the z- and *x*-axis directions are associated with the lower and intermediate inertia values, respectively.
**Algorithm 2.** Algorithm for the fine global registration of np point clouds using the standard semi-automatic alignment pipeline.**Require:**Point clouds {℘10, ℘20, …,℘np0}Labeling of point clouds from 1 to np in a clockwise direction
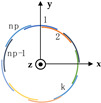
for h = 1 to np−1 do    Fix ℘h0 and leave ℘h+10 floating.    Manual selection of at least three corresponding points (i.e., fingertips, elbow, wrist bones, natural skin marks) on common areas of ℘h0 and ℘h+10    Rough alignment between the two adjacent point clouds using the 3-2-1 pairwise registration process.    Fine registration between the two adjacent point clouds by ICP algorithm.end forGlobal registration by ICP algorithm

Figure 11 illustrates, as an example, the results obtained for two cases. Each subplot in the figure displays a map of distances between the reference (clouds realigned with the manual method) and the alignment achieved with the proposed method. Additionally, a histogram is provided for each case, showing the frequencies and cumulative frequencies of the obtained discrepancies. The cumulative frequencies are specifically marked for distances of 5 mm and 10 mm. The two cases were selected to show the worst-case scenario (case#1) and the best-case scenario (case #2) obtained during the trials. As can be seen, even for case #1, 95% of the points have a distance lower than 5 mm, while, for case #2, the percentage rises to 98%.

Table 2 reports the results in terms of the means and standard deviation values of the same parameters obtained by analyzing all the cases. Most deviations were below values considered acceptable by the medical personnel involved in this research activity. In particular, an average of 96% of the points were found to have a distance of less than 5 mm compared with the model obtained with the conventional pipeline, while the average distance between the two clouds was less than 2 mm. The distinction between hand and arm was made to verify whether differences appear between almost uniform surfaces (like those of the arm) and surfaces characterized by a high degree of optical undercuts (for example, the fingers). The finger region poses the most significant challenges during the acquisition process, primarily due to self-undercuts, rapid slope variations of the surface, and the tendency for involuntary movements of the fingers. Even if the acquisition by the three RGB-D cameras is almost simultaneous for each scanner pose, the reconstruction of the overall limb anatomy requires the scanner to be moved around the arm. For this reason, any non-rigid limb movement during the acquisition process (which can last up to 50 s) effectively modifies the geometry to be reconstructed, resulting in data that cannot be perfectly realigned regardless of the methodology adopted.

Nevertheless, the data in Table 2 confirm that the two procedures (i.e., conventional manual registration and fully automated registration) provide almost indistinguishable results, since a relative point-by-point distance of less than 2 mm is considered negligible for soft anatomical tissues. Additionally, it is worth noting that the manual registration procedure requires an experienced operator with a point-cloud-processing background (thus, not medical staff) and about 20 min for a full 360° scan. On the other hand, the proposed prototype implementation of a fully automated procedure requires up to 30 min without human intervention. Specifically, the two procedures are differentiated by the time distribution of their two main steps. The gold-standard approach is time-consuming at the coarse alignment step since the operator must visually identify and select corresponding points on adjacent point clouds. The subsequent refinement using ICP takes less time due to optimized routines within commercial software due to the skilled and trained operator’s ability to visually identify corresponding landmarks. This visual identification results in an initial alignment of the point clouds, serving as a well-suited initialization for the subsequent optimization process. On the other hand, the developed approach is much faster during coarse initial alignment since identifying corresponding landmarks is automatic (neural network + forearm key-point detection) but is much slower during the convergence of the fine registration. The substantial advantage of a fully automated process, devoid of any operator intervention with specialized training, outweighs any potential increase in the time needed to achieve the outcome, making its effective use in clinical centers possible. Finally, it is essential to acknowledge that the developed procedure could benefit from an improved software implementation.

### 4.2. Improvements to the State of the Art

The performance of the proposed method was compared with algorithms that had been previously published in the literature and for which implementations were available. This choice was made to ensure a fair comparison among various techniques, as inaccuracies in implementing procedures developed by other authors could compromise the integrity of the comparison. Even minor errors might lead to cascading effects, resulting in significantly divergent outcomes, making meaningful algorithmic comparisons challenging. The overall accuracy of the procedure hinges on the precision of individual phases, including scanning, coarse registration, and fine registration. Given that the proposed method’s objectives involve generating accurate 3D models of upper limb anatomy through a low-cost handheld 3D scanner, this section of the study specifically compares the proposed method’s accuracy with the current state of the art for both coarse and fine registration. In particular:-The proposed algorithm for coarse registration is compared with the following:✓4PCS [24];✓Super4PCS [25];✓PCA [56];✓FPFH [57].-The proposed algorithm for fine registration is compared with the following:✓ICP point-to-plane [58];✓ICP trimmed (Tukeyloss) point-to-plane [59].

The comparison was carried out using the same five test cases and the corresponding gold standards presented in the previous section. Additionally, for assessing the outcomes produced by the compared fine-registration algorithms (specifically, ICP point-to-plane and ICP trimmed point-to-plane), the proposed coarse-registration method provides the initial solution.

Table 3 compares the mean distances and the standard deviation values of the analyzed coarse-registration methods with the corresponding gold standard. The parameters listed for each of the five methods are those for which the best results were obtained. Values for 4PCS [24] and Super4PCS [25] are not reported as the methods do not provide acceptable solutions. It is also evident that the proposed procedure, based on the realignment of specific anatomical features of the upper limb, allows for much more accurate initial alignment of 3D point clouds than other analyzed state-of-the-art methods.

Table 4 compares the mean distances and the standard deviation values of the analyzed fine-registration methods with the corresponding gold standard. Again, in this case, the parameters listed for each of the three methods are those for which the best results were obtained. By implementing the original function to be minimized, the proposed method achieves a notably higher level of accuracy in aligning point clouds compared with the methods examined in the current state of the art. Moreover, the low standard deviation values associated with the proposed method denote the promising robustness of the results obtainable by aligning the upper limb point clouds.

## 5. Conclusions

This research paper proposes a novel approach for automatically aligning and registering point clouds obtained by scanning upper limb anatomy with a depth-camera-based scanner. The methodology integrates information extracted from both 2D and 3D data. Specifically, a pre-trained neural network (NN) was exploited to automatically detect hand key points by processing RGB images captured by the depth cameras. Additionally, an automated algorithm was developed to identify the forearm’s skeleton line directly from the point cloud data. The combination of hand key points and forearm key points allows for better constraint of the coarse registration of point clouds obtained from different scanner poses. The automatic recognition of forearm features further enhances the methodology’s robustness, enabling it to handle a wide range of hand poses and even point clouds with limited overlap and noise.

Additionally, the coarse alignment step has proven effective in providing a reliable starting alignment for the subsequent fine-registration process, which is obtained through an innovative registration algorithm (inspired by ICP). To evaluate the proposed methodology, its results were analyzed using, as the reference, a gold-standard point cloud realignment pipeline conducted using a hybrid manual and automatic procedure within commercial software. The assessment demonstrated the effectiveness of the proposed approach in achieving accurate registration results. Most deviations were found to be below acceptable values, with approximately 96% of the points having a distance of less than 5 mm compared with the model obtained with the conventional pipeline (with an average distance value lower than 2 mm). This makes the proposed method a feasible alternative to manual registration, which also requires the intervention of a skilled operator, introducing some subjectivity in identifying common anatomical landmarks. Finally, the improvements over the state of the art were verified by comparing the proposed method with existing published methods for which implementations were accessible.

The main aim of the proposed methodology is to contribute to developing an effective and personalized upper limb rehabilitation framework. In this context, assembling a compact hand-held scanner and an automated procedure for registering partially overlapped and noisy point clouds is of utmost importance to allow the procedure to be completed in clinics without the intervention of reverse engineering experts. Although the approach has demonstrated encouraging outcomes, it is essential to conduct additional validation and testing using more comprehensive and varied datasets. This is necessary to thoroughly evaluate its performance in various scenarios, especially in clinical environments involving actual patients exhibiting potential movement disorders. Subsequent clinical validation will encompass the three-dimensional reconstruction of upper limb anatomy for individuals experiencing diminished mobility in their upper limbs. This includes cases related to patients who have undergone sports injuries or suffered strokes. The main limitation of the proposed approach is given by the 3D scanner, which requires multiple placements around the arm to complete the data acquisition process. This could be an issue in the case of patients not being able to maintain the arm in the same configuration for the time required to complete the scanning process (as in the case of fingers or wrist movements). However, the choice of the proposed scanner and scanning strategy was oriented towards defining a tradeoff between cost and accurate results. Future design efforts should be towards the definition of alternatives to mitigate or remove this limitation.

## Figures and Tables

**Figure 1 sensors-23-07841-f001:**
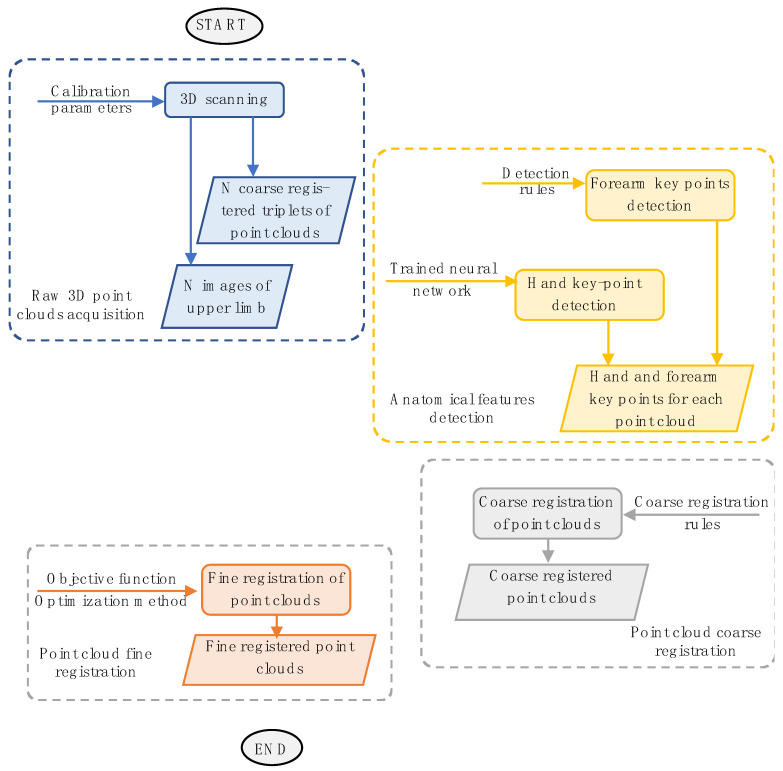
Flow-chart of the proposed method.

**Figure 2 sensors-23-07841-f002:**
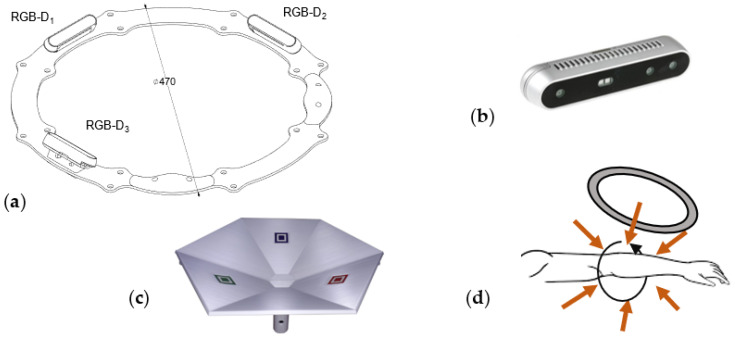
3D scanner architecture: (**a**) arrangement of the three depth cameras on the circular frame; (**b**) picture of a single sensor; (**c**) calibration specimen; (**d**) use of the scanner for the arm’s acquisition.

**Figure 3 sensors-23-07841-f003:**
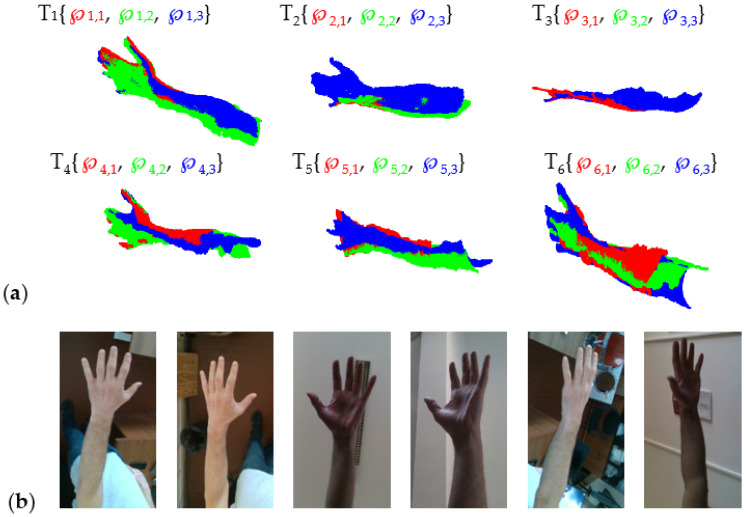
The results of the raw-data acquisition process: (**a**) N triplets of coarse registered point clouds; (**b**) some of the 3 × N images of upper limb anatomy from different orientations.

**Figure 4 sensors-23-07841-f004:**
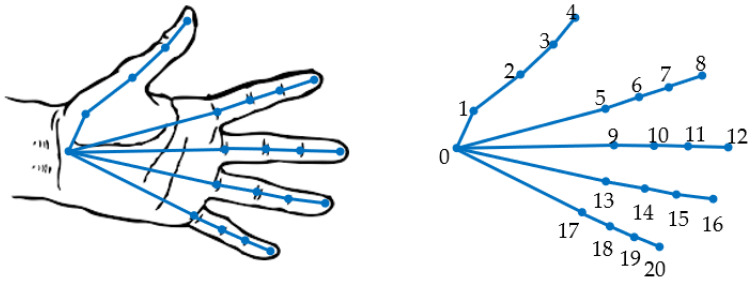
Hand key-point numbering in the NN.

**Figure 5 sensors-23-07841-f005:**
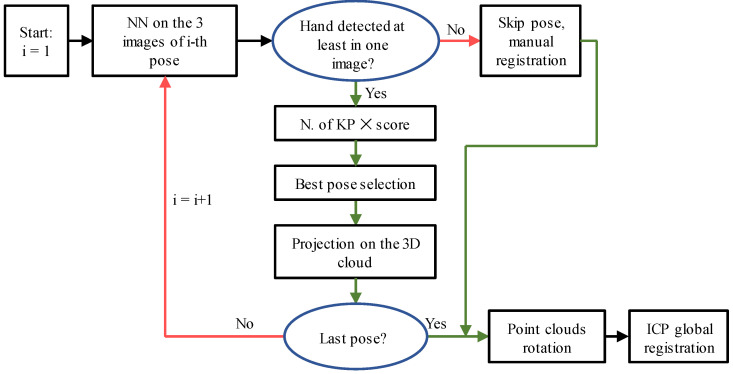
Automatic selection of successful key points.

**Figure 6 sensors-23-07841-f006:**
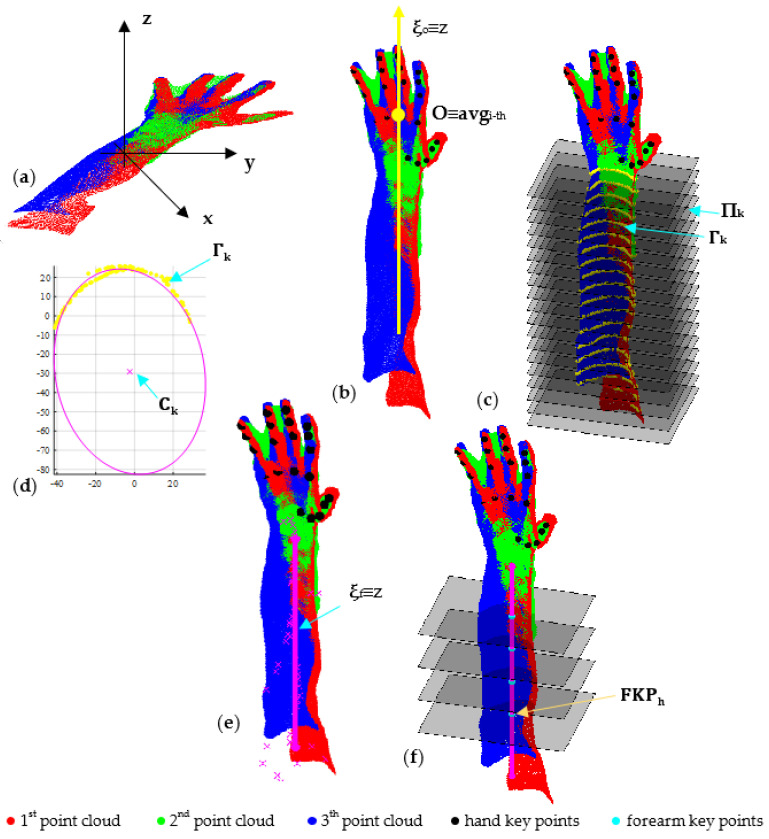
Key steps of the automatic process for forearm key-point detection: (**a**) the i−th triplet T_i_ corresponding to the *i*-th scanner pose; (**b**) alignment of T_i_ so that ξo coincides with the *z*-axis of the global reference system; (**c**) intersecting points Γ_k_ between T_i_ and the plane Π_k_ perpendicular to T_i_; (**d**) ellipse approximating Γ_k_ with a center at C_k_; (**e**) line approximating the center C_k_; (**f**) FKP_h_ identification.

**Figure 7 sensors-23-07841-f007:**
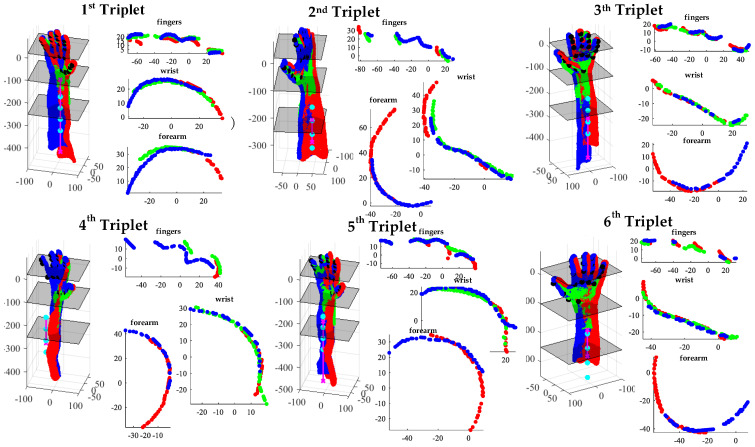
The results of the automatic registration performed after the acquisition stage and key−point detection for the six triplets reported in Figure 3.

**Figure 8 sensors-23-07841-f008:**
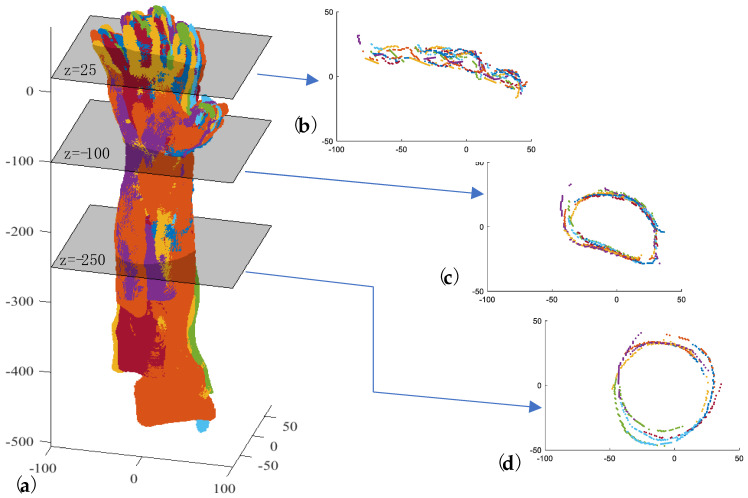
(**a**) The results of the coarse registration of the six triplets reported in Figure 3 Each color represents a sub−pointcloud coming from a depth sensor. (**b**) cross-section at z = 25 mm; (**c**) cross-section at z = 100 mm; (**d**) cross-section at z = 250 mm.

**Figure 9 sensors-23-07841-f009:**
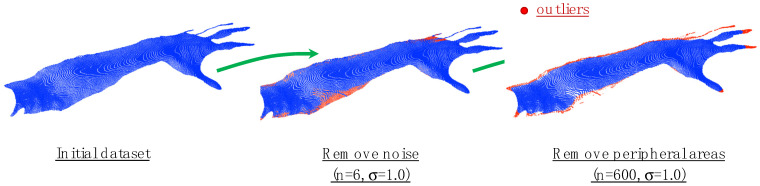
The results of the fine registration of the six triplets reported in Figure 3.

**Figure 10 sensors-23-07841-f010:**
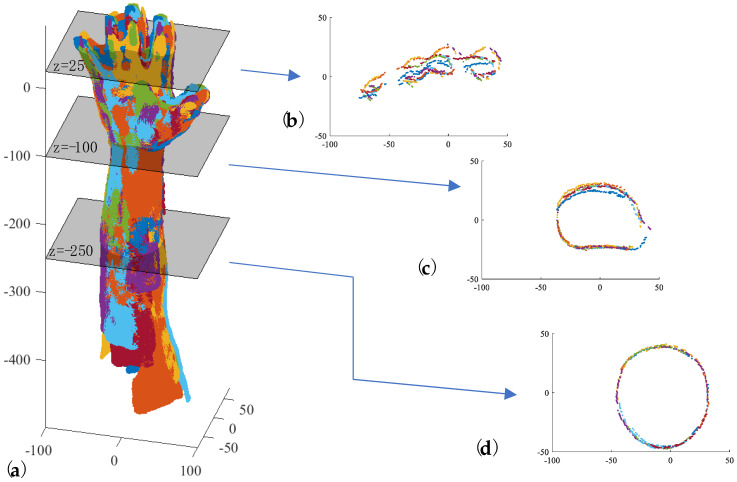
(**a**) The results of the fine registration of the six triplets reported in Figure 3. Each color represents a sub−pointcloud coming from a depth sensor. (**b**) cross−section at z = 25 mm; (**c**) cross-section at z = 100 mm; (**d**) cross-section at z = 250 mm.

**Figure 11 sensors-23-07841-f011:**
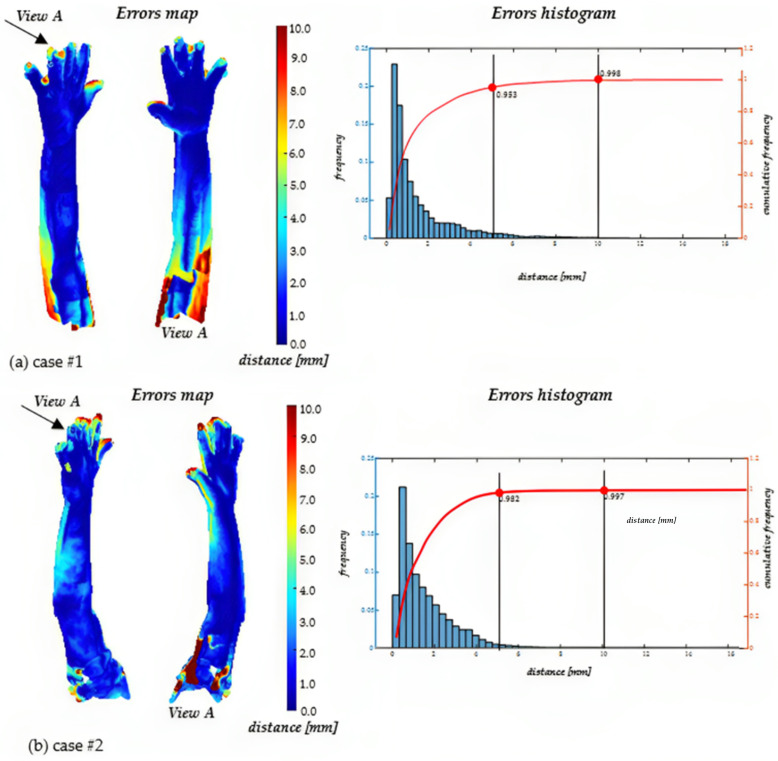
Examples of the results obtained.

**Table 1 sensors-23-07841-t001:** List of parameters and corresponding values for each step of the proposed methodology.

Step	Parameter	Values
Hand key-point detection	Detection score threshold	0.5
Forearm key-point detection	*H_FKP_*: heights below the barycenter to define forearm key points	−200 mm:−50 mm:−350 mm
Coarse registration of point clouds	*w_H_*: Horn’s quaternion-based method weights [54]	1
Fine registration of point clouds	SLSQP Voxel grid	0.75 mm
	*Toll*: distance threshold value (Equations (3) and (5))	1.5 mm
	SLSQP Stopping criteria	$\left\|{\frac{1}{f_{1}f_{2}}}^{iter}-{\frac{1}{f_{1}f_{2}}}^{iter-1}\right\|<10^{-4}$
	N_iter_ex_ (Algorithm 1)	9

**Table 2 sensors-23-07841-t002:** Mean distances and standard deviation values from the gold standard.

Parameter		Hand	Arm	Total
% points dist < 3 mm	mean	0.884	0.859	0.870
	std	0.088	0.073	0.014
% points dist < 6 mm	mean	0.979	0.975	0.976
	std	0.021	0.018	0.010
Dist, mm	mean	1.429	1.634	1.522
	std	0.572	0.363	0.026

**Table 3 sensors-23-07841-t003:** Comparison of mean distances and standard deviation values from coarse-registration methods and the gold standard.

Method	Parameters	Distances [mm]					
		Total		Hand		Arm	
		Mean	Std	Mean	Std	Mean	Std
Proposed method	See Table 1	2.440	0.742	2.871	1.011	2.175	0.725
4PCS [24]	d = 5, *n* = 200, t = 1000	--	--	--	--	--	--
Super4PCS [25]	d = 5, *n* = 200, t = 1000	--	--	--	--	--	--
PCA	*n* = 3	9.616	0.446	6.233	1.162	11.638	0.160
FPFH	*n* = 3	5.137	1.012	4.154	0.911	5.722	1.031

**Table 4 sensors-23-07841-t004:** Comparison of mean distances and standard deviation values from fine-registration methods and the gold standard.

Method	Parameters	Distances [mm]					
		Total		Hand		Arm	
		Mean	Std	Mean	Std	Mean	Std
Proposed method	See Table 1	1.429	0.572	1.634	0.363	1.522	0.026
ICP point-to-plane	d = 0.75, knn = 30	4.425	3.733	4.619	4.027	4.315	3.575
ICP trimmed (Tukeyloss) point-to-plane	d = 0.75, knn = 30, sigma = 1.5	2.964	1.079	3.095	1.274	2.891	0.972

## Data Availability

The data that support the findings of this study are available from the corresponding author, E.G., upon reasonable request.

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
