# Peer review of "Automatic Multiview Alignment of RGB-D Range Maps of Upper Limb Anatomy"

_sensors, 2023, doi:10.3390/s23187841_

Round 1

Reviewer 1 Report

- All results should briefly added to Abstract. 

- Introduction should be enriched with more recent references. 

- A deep comparison with other techniques is required. 

Reviewer 2 Report

The authors present an innovative methodology for automatically aligning and registering point clouds of upper limb anatomy, which they suggest could greatly facilitate upper limb rehabilitation.

The methodological transparency of the manuscript is commendable. The authors provide substantial details about their neural network and alignment algorithm, which improves the replicability of their work. The manuscript offers robust validation of the proposed method through comprehensive testing and statistical analysis, which aids in illustrating the effectiveness of their approach.

The manuscript's structure and flow of information are coherent and logical. The use of a neural network is appropriately discussed in conjunction with key point extraction. The use of visual aids such as device setup images, key point examples, and visualizations of test results greatly enhances the understanding of the proposed method.

However, there are a few areas that could be improved upon:

Abstract: The abstract could better highlight the testing and validation strategies used and could offer a brief overview of the statistical findings. While the manuscript does provide timing for each process, it does not explicitly discuss the differences in time taken by the manual process versus the automated process. Offering such a comparison could strengthen the authors' claim of efficiency for their proposed method.

Related work: This section offers a good overview of previous work in this area. It could be helpful, however, to directly highlight how this study's methodology differs from or improves upon the methods used in the studies mentioned.

Results and Discussion: From a critical perspective, one potential weakness in this section may pertain to the limited experimental sample; utilizing only healthy volunteers for the study could overlook potential circumstances under conditions of illness. The study uses a very limited sample size of only five healthy volunteers. This not only raises concerns about the generalizability of the results but also limits the study's application for patients with upper limb pathologies. In future work, a larger and more diverse sample size, including patients with various upper limb pathologies, should be considered. Consider explaining the impact of outliers on the overall results. When discussing the time comparison, it would be helpful to have data on the distribution of time it takes for manual and automatic process rather than just providing an average.  Moreover, while the paper does touch on some pros and cons of both the manual and automatic methods, it doesn't delve into these advantages and disadvantages in depth. A further discussion and comparison of the pros and cons of these methods may be more beneficial to the reader. 

Lack of Comparative Analysis: While the authors compare their approach to a manual method, there is a lack of comparison with other existing automatic methods. It would be more convincing if the authors compared their methodology with other state-of-the-art automatic point cloud alignment methods.

Software Details: The authors mentioned that custom software was developed in MATLAB, incorporating functions written in Python. More detailed information regarding the software implementation should be included to ensure reproducibility of the results.

Performance Metrics: The manuscript primarily uses the distance of points as the performance metric. Incorporation of other metrics, like precision, recall, F1-score etc., would provide a more comprehensive evaluation of the proposed method.

Timing of Procedures: The manuscript states that the manual method takes about 20 minutes, while the automatic procedure requires up to 30 minutes. It would be useful to clarify why the automated method takes longer and discuss how this might be improved in future implementations.

Future Improvements: The conclusion lacks a robust discussion of possible future improvements or next steps. For example,although the results of the proposed methodology show promising potential for clinical applications, the study lacks validation with real-world clinical data. Mentioning areas of potential improvement and the scope for future work will provide the paper with a more solid ending.

Could be improved.

Author Response

Reviewer #2

  • The authors present an innovative methodology for automatically aligning and registering point clouds of upper limb anatomy, which they suggest could greatly facilitate upper limb rehabilitation.

The methodological transparency of the manuscript is commendable. The authors provide substantial details about their neural network and alignment algorithm, which improves the replicability of their work. The manuscript offers robust validation of the proposed method through comprehensive testing and statistical analysis, which aids in illustrating the effectiveness of their approach.

The manuscript's structure and flow of information are coherent and logical. The use of a neural network is appropriately discussed in conjunction with key point extraction. The use of visual aids such as device setup images, key point examples, and visualizations of test results greatly enhances the understanding of the proposed method.

Response: The Authors wish to thank the Reviewer for the positive comments about the paper.

However, there are a few areas that could be improved upon:

  • Abstract: The abstract could better highlight the testing and validation strategies used and could offer a brief overview of the statistical findings. While the manuscript does provide timing for each process, it does not explicitly discuss the differences in time taken by the manual process versus the automated process. Offering such a comparison could strengthen the authors' claim of efficiency for their proposed method.

Response: The authors slightly modified the abstract by better describing the validation strategies and providing some details of the statistical findings.

Concerning time issues, the authors have discussed differences in time of the manual conventional approach for the automated process in the discussion section. While the observed differences might not hold significant importance, the author believes that using time as a basis for comparison merely provides a preliminary indication. This viewpoint stems from the considerable benefits of a completely automated process that eliminates the need for operator involvement and specialized training, thus improving efficiency. These advantages are deemed more valuable than any potential increase in the time required to achieve the outcome. It should also be considered that iterative processes, such as optimization algorithms, benefit significantly from having a good starting point, which is easier to achieve if a skilled and trained operator manually selects corresponding points for the initial coarse alignment. The operator provides the algorithm with a sensible starting configuration, which can help it converge more quickly and accurately toward a solution. Finally, the time of the proposed procedure could drastically decrease through better software implementation, which is beyond the aim of the present paper. This issue has been clarified in the discussion section.

  • Related work: This section offers a good overview of previous work in this area. It could be helpful, however, to directly highlight how this study's methodology differs from or improves upon the methods used in the studies mentioned.

Response: According to this reviewer's comment, sentences in different sections of the paper have been added to "highlight how this study's methodology differs from or improves upon the methods used in the studies mentioned."

  • Results and Discussion: From a critical perspective, one potential weakness in this section may pertain to the limited experimental sample; utilizing only healthy volunteers for the study could overlook potential circumstances under conditions of illness. The study uses a very limited sample size of only five healthy volunteers. This not only raises concerns about the generalizability of the results but also limits the study's application for patients with upper limb pathologies. In future work, a larger and more diverse sample size, including patients with various upper limb pathologies, should be considered. Consider explaining the impact of outliers on the overall results. When discussing the time comparison, it would be helpful to have data on the distribution of time it takes for manual and automatic process rather than just providing an average.  Moreover, while the paper does touch on some pros and cons of both the manual and automatic methods, it doesn't delve into these advantages and disadvantages in depth. A further discussion and comparison of the pros and cons of these methods may be more beneficial to the reader. 

Response: The authors agree with the reviewer that no clinical validation has been presented in the present work. The research's final aim is to use the proposed approach in clinical scenarios with patients affected by upper limb pathologies (deriving from stroke, sports injuries or dystonia disorders). However, this is a preliminary step, and the authors aim to demonstrate the procedure's feasibility using healthy subjects. In the conclusions section, this aspect of future work has been outlined. Concerning the time considerations, it must be considered that most of the time (up to 90%) is spent in the pair-wise alignment with the operator that must manually select corresponding points between pairs of adjacent point clouds. The following registration refinement by ICP is a fast step since it is carried out with optimized routines of commercial software.

Regarding the impact of outliers on the overall results, they significantly affect the final aligned configuration. Therefore, according to this reviewer's statement, additional comments and a new figure have been introduced in section 3.3.

  • Lack of Comparative Analysis: While the authors compare their approach to a manual method, there is a lack of comparison with other existing automatic methods. It would be more convincing if the authors compared their methodology with other state-of-the-art automatic point cloud alignment methods.

Response: According to this reviewer's comment, the performances of the proposed method were compared with algorithms published in the literature for which the implementation was available. This choice is because the correctness of implementing procedures developed by other authors could undermine a fair comparison between different techniques: even minor errors can have cascading effects and produce drastically different outcomes, making meaningful comparisons between algorithms challenging. Since the accuracy of the whole procedure depends on the accuracy of individual phases (scanning, coarse registration and fine registration), and the proposed method aims to create accurate 3D models of the upper limb anatomy using a low-cost handheld 3D scanner, comparison with the state-of-the-art are carried out for coarse and fine registration. In particular:

-           the proposed algorithm for coarse registration is compared with the following:

  • 4PCS;
  • Super4PCS;
  • PCA;
  •  

-           the proposed algorithm for fine registration is compared with the following:

  • ICP point-to-plane;
  • ICP trimmed (Tukeyloss) point-to-plane.

The results obtained were critically discussed in a new section (4.2).

  • Software Details: The authors mentioned that custom software was developed in MATLAB, incorporating functions written in Python. More detailed information regarding the software implementation should be included to ensure reproducibility of the results.

Response: The authors understand the reviewer's point of view. However, enough details have been provided about the procedure to ensure reproducibility. The adopted neural network is detailed in section 3.2.1; the forearm key points detection is detailed in section 3.2.2; the fine registration process is described in section 3.4 by providing objective functions, pseudo-code and a list of parameters (Table 2).

Since some commercial and confidentiality constraints exist, we aim to balance sharing sufficient information to uphold the scientific process's integrity while respecting the proposed methodology's proprietary nature. However, the authors can actively explore avenues to release more details to single researchers within these constraints.

  • Performance Metrics: The manuscript primarily uses the distance of points as the performance metric. Incorporation of other metrics, like precision, recall, F1-score etc., would provide a more comprehensive evaluation of the proposed method.

The precision P0, recall R0 and F1-score F0 are defined in literature as follows:

where TP, FN, and FP are the number of true positive points, false negative points, and false positive points. These performance metrics are typically used to measure the performances of recognition algorithms. Applying these metrics to the global registration of point clouds acquired with different cameras at different times is not straightforward since an arbitrary method to identify true positive points, false negative points, and false positive points should be used.

  • Timing of Procedures: The manuscript states that the manual method takes about 20 minutes, while the automatic procedure requires up to 30 minutes. It would be useful to clarify why the automated method takes longer and discuss how this might be improved in future implementations.

Response: The two procedures differentiate for the time distribution between the two main steps. The conventional approach (manual selection of corresponding points for the coarse alignment + ICP algorithm for fine registration using commercial software) is time-consuming for the first process since an operator must visually identify and select corresponding points on adjacent points clouds for initial coarse registration. The subsequent ICP refinement instead takes less time because it is carried out by commercial software starting from a good initial solution. On the contrary, the developed approach is much faster in the coarse initial alignment since the identification of corresponding landmarks is demanded to an automatic procedure (neural network + forearm key points detection), but is much slower in the convergence of the fine registration. This is undoubtedly due to our algorithmic implementation, which could certainly be optimized but could also be ascribed to a worst key points identification, which an operator does not supervise. These considerations have been better highlighted in the revised version of the manuscript.

  • Future Improvements: The conclusion lacks a robust discussion of possible future improvements or next steps. For example,although the results of the proposed methodology show promising potential for clinical applications, the study lacks validation with real-world clinical data. Mentioning areas of potential improvement and the scope for future work will provide the paper with a more solid ending.

Response: The authors agree with the reviewer that real-world clinical validation is missing and should be considered in future works. The present paper aims to verify the feasibility of the approach in the case of healthy subjects, able to maintain the arm in an extension condition for about 30 seconds without introducing any other variable. Of course, working within clinical scenarios would present the need to consider patients with reduced upper limb mobility (for example, patients from sports injuries or strokes) or patients with uncontrolled arm movements, such as those affected by dystonia disorders. However, in the authors' opinion, the main limitation of the proposed approach is given by the 3D scanner, which requires multiple placements around the arm to complete the data acquisition process. This could be an issue in the case of patients not being able to maintain the arm in the same configuration for the time required to complete the scanning process (for example, in the case of fingers or wrist movements). However, the scanner and scanning strategy choice was oriented towards defining a tradeoff between cost and result accuracy. These considerations have been further added in the conclusion section.

Reviewer 3 Report

The authors describe a method to assess forearm superficial anatomy in vivo  by means of an handheld 3D scanner that uses established mathematical algorithms, commercial software and python libraries.

Introduction, methods and discussion are well structured and clear. There is no comparison of the assessed method with some kind of a "gold standard".

Author Response

Reviewer #3

The authors describe a method to assess forearm superficial anatomy in vivo by means of a handheld 3D scanner that uses established mathematical algorithms, commercial software and python libraries.

  • Introduction, methods and discussion are well structured and clear.

Response: The Authors wish to thank the Reviewer for the positive comments about the paper.

  • There is no comparison of the assessed method with some kind of a "gold standard".

Response: According to this reviewer's comment, the performances of the proposed method were compared with algorithms published in the literature for which the implementation was available. This choice is because the correctness of implementing procedures developed by other authors could undermine a fair comparison between different techniques: even minor errors can have cascading effects and produce drastically different outcomes, making meaningful comparisons between algorithms challenging. Since the accuracy of the whole procedure depends on the accuracy of individual phases (scanning, coarse registration and fine registration), and the proposed method aims to create accurate 3D models of the upper limb anatomy using a low-cost handheld 3D scanner, comparison with the state-of-the-art are carried out for coarse and fine registration. In particular:

-           the proposed algorithm for coarse registration is compared with the following:

  • 4PCS;
  • Super4PCS;
  • PCA;
  • FPFH.

-           the proposed algorithm for fine registration is compared with the following:

  • ICP point-to-plane;
  • ICP trimmed (Tukeyloss) point-to-plane.

The results obtained were critically discussed in a new section (4.2).

Reviewer 4 Report

1) What is the effective range of the camera's depth sensing capabilities? Will it cover extremely close or distant objects, which could be a limitation in this application.

2) Like many depth sensors, the D415 might produce some level of noise in the depth data, leading to minor inaccuracies or artifacts in the captured 3D information. How did the authors reduce the noise generated?

3) Depth cameras can be sensitive to lighting conditions. Overly bright or low-light environments might affect the camera's performance, leading to decreased accuracy or missing depth data. What are the lighting condition used?

4) While MediaPipe's Hand Landmarker model is generally accurate, it might not always provide perfect results, especially in challenging conditions such as complex hand poses, occlusions, and varying lighting conditions. The accuracy might be affected when fingers are close together or when hands are partially obstructed. How has the authors overcome these problems?

5) The model is specialized for hand landmark detection, which means it might not perform well in scenarios that require more comprehensive hand tracking, such as tracking the entire hand's movement in 3D space which is the real-time application

6) A much more comprehensive validation on different datasets with different key points are required to authenticate the study’s effectiveness.

7) What are the future scope of this study? How to improve the design?

Minor editing of English language required.

Author Response

Reviewer #4

  • What is the effective range of the camera's depth sensing capabilities? Will it cover extremely close or distant objects, which could be a limitation in this application.

Response: The Intel RealSense D415 has a wide acquisition depth sensing, which ranges from about 160 mm to 10000 mm. However, Carfagni et al. have demonstrated that the maximum accuracy of the depth camera is obtained for close distances (from 150 mm to 500 mm) as the 3D reconstruction error in this range can be below 1 mm [Carfagni, M., Furferi, R., Governi, L., Santarelli, C., Servi, M., Uccheddu, F., Volpe, Y.: Metrological and Critical Characterization of the Intel D 415 Stereo Depth Camera. Sensors-Basel 19, (2019)]. The three depth cameras have been arranged in the present work to acquire the target shape in the 250 – 450 mm range. This information has been added to the revised version of the manuscript.

  • Like many depth sensors, the D415 might produce some level of noise in the depth data, leading to minor inaccuracies or artifacts in the captured 3D information. How did the authors reduce the noise generated?

Response: The authors agree with the reviewer. The depth camera can be considered a low-cost scanner, and clearly, the noise level is much higher than high-end structured light scanners. In this specific application, we have identified two kinds of inaccuracies in the acquired datasets: a random noise, identifiable in the overall point cloud, and a systematic error, characterizing the peripheral areas of the arm and the hand. Both these have been reduced by applying a SOR filter during the fine registration. All these considerations and a new figure (Figure 9) have been introduced in section 3.3 of the paper to describe better and summarize the procedure.

  • Depth cameras can be sensitive to lighting conditions. Overly bright or low-light environments might affect the camera's performance, leading to decreased accuracy or missing depth data. What are the lighting condition used?

Response: The 3D scanner has been developed to be used in clinical scenarios. For this reason, tests have been carried out in room ambient lighting conditions. This consideration has been added to the revised version of the manuscript.

  • While MediaPipe's Hand Landmarker model is generally accurate, it might not always provide perfect results, especially in challenging conditions such as complex hand poses, occlusions, and varying lighting conditions. The accuracy might be affected when fingers are close together or when hands are partially obstructed. How has the authors overcome these problems?

Response: The authors agree with the reviewer’s consideration. Finger occlusion due to specific hand or finger poses represents a limit for correctly identifying hand key points by the neural network. However, it is worth noting that the coarse alignment requires a minimum of three key points, and the developed scanner consists of three depth cameras, thus introducing a redundancy that increases the possibility that at least three key points are identified by at least one of the three sensors. Clearly, the greater the key points identified, the more precise the coarse alignment is, thus speeding up the subsequent registration process. If some hand poses lead to the detection of less than three hand key points, the automatic coarse alignment fails, and an operator must intervene to align the relative scan to the previous ones. These issues have been better clarified in section 3.2.1 of the manuscript.

  • The model is specialized for hand landmark detection, which means it might not perform well in scenarios that require more comprehensive hand tracking, such as tracking the entire hand's movement in 3D space which is the real-time application.

Response: The adopted model only acquires the upper limb anatomy in steady condition. The aim is to reconstruct the 3D model of the upper limb with the least possible intervention of skilled operators (to create a framework that can be used within clinical scenarios) and exploit low-cost hardware. The hand tracking is out of the scope of the present research activity. The 3D model could be effectively used to design bespoke appliances (for example, controllers) to enhance rehabilitation tasks.

  • A much more comprehensive validation on different datasets with different key points are required to authenticate the study’s effectiveness.

Response: The authors concur with the reviewer's observation regarding the usefulness of adding more datasets to gain a more robust validation. However, this work aimed to verify the feasibility of the proposed approach. For this reason, only healthy subjects were considered since they could maintain the arm in an extended configuration for the time required to complete the scanning process.

  • What are the future scope of this study? How to improve the design?

Response: The future scope of this study is to validate the proposed methodology in clinical scenarios. Future clinical validation will consider the 3D reconstruction of the upper limb anatomy of individuals with reduced upper limb mobility (as that related to patients who suffered from sports injuries or stroke). In the authors' opinion, the main limitation of the proposed approach is given by the 3D scanner, which requires multiple placements around the arm to complete the data acquisition process. This could be an issue in the case of patients not being able to maintain the arm in the same configuration for the time required to complete the scanning process (as in the case of fingers or wrist movements). However, the choice of the proposed scanner and the scanning strategy was oriented towards defining a tradeoff between cost and result accuracy. Future design efforts should be oriented towards defining alternatives to mitigate or remove this limitation. These considerations have been highlighted in the conclusions section.

Author Response

Reviewer #5

This paper describes a novel procedure for creating 3D model of the arm, exploiting both hardware and software solution designed by the authors. The prototype has been tested on real forearm and the resulting 3D models have been compared to a manual procedure. The overall quality of the paper is really high, state of the art is exhaustive, the arm reconstruction algorithm is clearly presented and result are quite encouraging (computational times must reduced in future).

Response: The Authors wish to thank the Reviewer for the positive comments about the paper.

Therefore, I have only some minor comments:

  • Figure 6: Please provide also in the caption a brief description of the key point detection process describing each one of the subfigures.

Response: According to this reviewer’s comment, the caption of figure 6 has been modified.

  • Based on the captions of figures 7 and 9 it seems that its the same figure. Please clarify.

Response: According to this reviewer’s comment, the caption of figure 7 has been modified.

  • Figure 10: there is no reason the use two different x-scale between the histogram. Please report the error histogram of case 2 in the same scale of case 1.

Response: According to this reviewer’s comment, figure 10 (figure 11 in the modified version of the paper) has been modified.

Round 2

Reviewer 2 Report

The authors have provided satisfactory responses to the reviewer's comments. Recommend publication as is.